# PRISM-EE: A Peer-Federated Framework for Cost-Aware Large Language Model Evaluation

## Abstract

Large Language Model evaluation faces three critical problems: static benchmarks suffer from data contamination, human-judged systems have systematic biases, and most importantly, both ignore cost. The key factor determining real-world deployment decisions. We introduce PRISM-EE (Peer-Reviewed Intelligence Scoring Methodology with Economic Evaluation), a peer-federated framework where AI models evaluate each other through specialized roles: competitors solve problems, content creators design challenges, and judges evaluate solutions. This approach generates fresh content dynamically while reducing human bias. PRISM-EE evaluates models on dual tracks: raw performance and cost efficiency. Using Swiss-style pairing, we achieve stable ratings in 25-30 matches with ±18 Elo precision, compared to 100+ matches required by existing systems. We tested 48 models across clinical reasoning, mathematics, and programming domains. Results reveal dramatic cost variations invisible to traditional benchmarks: substantial efficiency gaps between models with similar capabilities, with some models delivering 97% of top performance at just 0.16% of the cost. PRISM-EE achieves 89% judge agreement compared to 72% for human evaluators, with gaming resistance through cross-provider validation and transparent logging. The framework includes a comprehensive governance system ensuring fair evaluation opportunity for all models regardless of provider size. Our open-source framework makes economic efficiency a primary evaluation criterion, enabling better deployment decisions where both performance and cost matter.

## 1 Introduction

Current AI model evaluation suffers from three fundamental flaws that render it inadequate for real-world deployment decisions. First, static benchmarks like MMLU and MedQA have been compromised by data contamination, with 50-80% of improvements coming from memorization rather than genuine capability gains (Hendrycks et al., 2020). Second, human-judged systems like LM-SYS Chatbot Arena exhibit systematic bias, where major providers receive 62.8% of evaluation data while open models get only 29.7% (Singh et al., 2025). Third, and most critically, existing evaluations completely ignore economic efficiency—a critical factor in real-world deployment decisions.

We introduce PRISM-EE (Peer-Reviewed Intelligence Scoring Methodology with Economic Evaluation), a peer-federated framework that addresses these limitations through a self-regulating ecosystem where AI models evaluate each other (Achiam et al., 2023). The system employs three specialized roles: competitors solve problems, content creators design challenges, and judges evaluate solutions (Lightman et al., 2023). Swiss-style pairing (FIDE, 2018) ensures equal evaluation opportunity for all models, while dual-track ratings measure both raw performance and cost-efficiency.

Our evaluation of 48 models across clinical reasoning, mathematics, and programming domains reveals dramatic cost variations invisible to traditional benchmarks. Some models deliver 97% of top performance at just 0.16% of the cost, with efficiency gaps reaching 641× between similar-performing models. The framework achieves 89% judge agreement compared to 72% for human evaluators (Chiang et al., 2024), while converging to stable ratings in 25-30 matches with ±18 Elo precision.

This work transforms AI evaluation into a functional marketplace where economic efficiency becomes a primary evaluation criterion, enabling better deployment decisions where both performance and cost matter.

## 1.1 KEY CONTRIBUTIONS

Our work makes six key contributions to AI evaluation:

**First cost-aware evaluation framework** that integrates economic efficiency as a primary criterion.

**Peer-federated methodology** where AI models evaluate each other through specialized roles.

**Fair opportunity guarantee** via Swiss-style pairing that ensures equal evaluation for all models.

**Framework reliability** explaining consistent model generation hierarchy across providers.

**Gaming resistance mechanisms** with cross-provider validation and cryptographic audit trails.

**Open-source implementation** with complete reproducibility and validated templates.

## 2 METHODOLOGY

### 2.1 SYSTEM ARCHITECTURE

PRISM-EE implements a peer-federated evaluation ecosystem with three distinct model roles:

**Competitors:** Models that solve problems generated by content creators. They receive challenges and provide responses that are evaluated by judges.

**Content Creators:** Models selected from Strata 4 (1520+ Elo) that generate case scenarios and questions for evaluation (Wang et al., 2022). Content must be validated by peer models before use, with penalties applied for low-quality contributions. This dynamic content generation approach (Kiela et al., 2021) ensures fresh evaluation challenges that prevent benchmark staleness.

**Judges:** High-performing models from Strata 4 (1520+ Elo) that evaluate competitor responses. They provide structured verdicts and are weighted based on their own performance ratings. Complete anonymity is maintained: case creators, question creators, and judges are selected independently from Strata 4 without knowledge of each other's identities. Penalties are applied for wrongful behavior including invalid verdicts, inconsistent evaluations, or gaming attempts.

### 2.2 GOVERNANCE FRAMEWORK

PRISM-EE operates as a peer-federated evaluation ecosystem where AI models evaluate each other, creating unique governance challenges that traditional evaluation systems never faced. The framework must address the critical trust problem of "who watches the watchers" when the evaluators themselves are AI models, manage economic stakes worth millions in deployment decisions where 641× efficiency gaps directly impact business outcomes, and scale to 100+ models while preventing sophisticated collusion attempts.

The governance framework operates on four interconnected layers: (1) **Technical Governance** with automated monitoring, cryptographic audit trails, and cross-provider validation; (2) **Community Governance** through a diverse governing council with transparent voting; (3) **Content Governance** ensuring quality control through peer review and expert validation; and (4) **Meta-Governance** managing framework evolution and long-term sustainability.

**Comprehensive Implementation Details**: A detailed governance framework addressing bootstrap policies, content governance, regular review cycles, scalability management, quality assurance protocols, and all operational procedures is provided in **Appendix A**. It is encouraged to examine the complete governance implementation for full technical and procedural details.

## 2.3 MATHEMATICAL FOUNDATIONS

A detailed mathematical framework explaining all formulas, parameter selection rationale, and implementation details is provided in **Appendix B**. The following sections summarize the core mathematical components.

### 2.3.1 DUAL-TRACK ELO RATING SYSTEM

The framework maintains two parallel rating systems (Elo & Sloan, 1978):

$$R_{\text{raw}}^{(t+1)} = R_{\text{raw}}^{(t)} + K \cdot (S_{\text{raw}} - E_{\text{raw}}) \quad (1)$$

$$R_{\text{cost}}^{(t+1)} = R_{\text{cost}}^{(t)} + K \cdot (S_{\text{adj}} - E_{\text{cost}}) \quad (2)$$

Where the cost-adjusted score incorporates efficiency weighting:

$$\text{eff}_A = \frac{e^{-C_A/\tau_c}}{e^{-C_A/\tau_c} + e^{-C_B/\tau_c}} \quad (3)$$

$$S_{\text{adj},A} = \frac{S_{\text{raw},A} \cdot \text{eff}_A}{S_{\text{raw},A} \cdot \text{eff}_A + S_{\text{raw},B} \cdot \text{eff}_B} \quad (4)$$

### 2.3.2 SWISS PAIRING ALGORITHM

Player 1 Selection (Inverse Match-Count Weighting):

$$P(m_i) = \frac{1/(1 + n_i)}{\sum_{j=1}^{N} 1/(1 + n_j)} \quad (5)$$

Player 2 Selection (Elo Stratum Matching) (Bradley & Terry, 1952):

$$S_\Delta(m_A) = \{m_j \in M \setminus \{m_A\} : \quad (6)$$

$$|R_j^{\text{cost}} - R_A^{\text{cost}}| \le 50\} \quad (7)$$

### 2.3.3 JUDGE WEIGHTING SYSTEM

Valid judge votes are weighted based on performance:

$$w_k = \frac{e^{R_k^{\text{raw}}/\tau}}{\sum_{j \in J_{\text{valid}}} e^{R_j^{\text{raw}}/\tau}} \quad (8)$$

### 2.3.4 COST CALCULATION METHODOLOGY

For each match, we track both input and output token costs to calculate(Bai et al., 2022) the total

cost per model:

$$C_A = c_{\text{in}} \cdot t_{\text{in},A} + c_{\text{out}} \cdot t_{\text{out},A} \quad (9)$$

Where $c_{\text{in}}$ and $c_{\text{out}}$ are the input and output token costs per million tokens, and $t_{\text{in},A}$ and $t_{\text{out},A}$ are the input and output token counts for model A.

We then calculate efficiency weights that favor more resource-efficient models:

$$\text{eff}_A = \frac{e^{-C_A/\tau_c}}{e^{-C_A/\tau_c} + e^{-C_B/\tau_c}} \quad (10)$$

With $\tau_c = 0.05$ determining the sensitivity to cost differences. The cost-adjusted score combines performance and efficiency:

$$S_A^{\text{adj}} = \frac{S_A^{\text{raw}} \cdot \text{eff}_A}{S_A^{\text{raw}} \cdot \text{eff}_A + S_B^{\text{raw}} \cdot \text{eff}_B} \quad (11)$$

This approach creates a natural trade-off: models using more resources must show proportionally better performance to maintain high cost-adjusted ratings.

### 2.3.5 JUDGE PENALTY SYSTEM

Judges providing invalid or low-quality evaluations receive penalties:

$$R_j^{\text{raw}} \leftarrow R_j^{\text{raw}} - P_{\text{judge}} \quad (12)$$

$$R_j^{\text{cost}} \leftarrow R_j^{\text{cost}} - P_{\text{judge}} \quad (13)$$

Where $P_{\text{judge}} = 10$ Elo points are deducted from both raw and cost-adjusted ratings for judges who fail to provide properly formatted verdicts or deliver inconsistent evaluations.

## 2.4 PARAMETER CONFIGURATION AND OPTIMIZATION

Key parameters were optimized through comprehensive grid search and cross-validation to ensure optimal system performance:

**Core Parameters:**

- $K = 16$: Adaptation factor optimized for 25-30 match convergence with $\pm 18$ Elo precision. Low K values (K=8) cause slow convergence requiring 80+ matches, while high values (K=32) create rating instability with $\pm 35$ Elo variance. K=16 provides optimal balance between adaptation speed and rating stability.

- $\tau_c = 0.05$: Cost sensitivity parameter calibrated for detecting substantial efficiency gaps. Low values ($\tau_c = 0.01$) make cost differences negligible, while high values ($\tau_c = 0.2$) overwhelm performance signals, causing 40% of models to rank purely by cost. The optimal value ensures cost differences are appropriately weighted without overwhelming performance considerations.

- $\tau = 300$: Judge weighting temperature optimized for 89% agreement rate. Low values ($\tau = 100$) create excessive judge hierarchy with top models dominating 90% of votes, while high values ($\tau = 500$) reduce weight differences to near-uniform distribution, dropping agreement to 72%. The optimal value balances judge influence with evaluation diversity.

- $\pm 50$ Elo: Swiss pairing tolerance ensuring competitive matches while maintaining sufficient opponent pool. Narrow tolerance ($\pm 25$) creates insufficient match opportunities for 15% of models, while wide tolerance ($\pm 100$) produces non-competitive matches with 85% win rates, reducing rating precision to $\pm 45$ Elo.

- $P_{\text{judge}} = 10$ Elo: Penalty amount for judges providing invalid or low-quality evaluations. The 10-point penalty is designed to make judges fall from Strata 4 (1520+ Elo) when they provide poor evaluations, effectively removing them from judging duties. This ensures only high-quality models serve as judges, as evidenced by GPT-o3's 29 penalties causing significant ranking drops.

# 3 EXPERIMENTS

## 3.1 EXPERIMENTAL SETUP

We evaluated 48 models across 7 providers (OpenAI, Google, Anthropic, Meta, xAI, Groq, Hugging Face) in three domains:

- Clinical Reasoning: 48 models, 1,198 matches (Primary evaluation) (Jin et al., 2019)
- Programming: 35 models, 372 matches (Domain extension)
- Mathematical Reasoning: 35 models, 157 matches (Domain extension)

Evaluation metrics include Raw Elo, Cost-adjusted Elo, convergence speed, and statistical robustness with confidence intervals and correlation analysis.

## 3.2 FAIR OPPORTUNITY GUARANTEE: SOLVING THE LMSYS PROBLEM

A critical limitation of existing evaluation systems like LMSYS Chatbot Arena is the systematic bias where models from major providers receive disproportionate evaluation opportunities (Singh et al., 2025). Studies reveal that major providers received 62.8% of arena data while open models got only 29.7% (Singh et al., 2025), creating a fundamental information asymmetry that undermines evaluation fairness.

PRISM-EE solves this through Swiss-style pairing with inverse match-count weighting that ensures equal evaluation opportunity for all models regardless of provider size or market position:

**Inverse Match-Count Weighting:**

$$P(m_i) = \frac{1/(1+n_i)}{\sum_{j=1}^{N} 1/(1+n_j)} \tag{14}$$

Where models with fewer completed matches receive higher selection probability, ensuring under-evaluated models are systematically prioritized.

**Evidence of Fair Opportunity:** Our clinical reasoning evaluation demonstrates perfect fairness— all 48 models received exactly 50 matches each (except GPT-4.1 mini with 46), regardless of provider:

## 3.3 PRIMARY RESULTS: CLINICAL REASONING

Our primary evaluation reveals dramatic divergence between raw performance and cost-adjusted rankings:

Table 1: Top 10 Models: Raw vs Cost-Adjusted Rankings

| Model | Raw Elo | Cost Elo | Cost ($) | Rank Change |
|---|---|---|---|---|
| Gemini 2.5 Pro | 1603.9 | 1561.9 | 37.75 | -4 |
| GPT-4.1 | 1601.2 | 1603.7 | 5.58 | 0 |
| GPT-4.1 mini | 1599.6 | 1605.7 | 1.11 | +2 |
| GPT-o4-mini | 1568.2 | 1569.9 | 4.50 | 0 |
| Qwen 3.2 235B | 1555.4 | 1564.8 | 0.059 | +1 |
| Grok 3 Mini Fast | 1537.7 | 1540.9 | 0.84 | 0 |
| GPT-o3-mini | 1533.7 | 1492.4 | 16.67 | -15 |
| Claude 3.7 Sonnet | 1530.5 | 1527.1 | 8.03 | -1 |
| Grok 3 | 1523.4 | 1523.2 | 8.05 | 0 |
| Qwen 3 32B | 1520.2 | 1537.7 | 0.071 | +3 |

## 3.4 CROSS-DOMAIN VALIDATION

Table 2: Cross-Domain Validation Results

| **Metric** | **Programming** | **Mathematical Reasoning** |
|---|---|---|
| Top Performer Performance (Elo) | Gemini 2.0 Flash 1586.2 | GPT-o4-mini 1571.2 |
| Most Efficient Cost per Evaluation | Qwen 3.2 235B $0.007 | Qwen 3 32B $0.019 |
| Models Evaluated | 35 | 35 |
| Total Matches | 372 | 157 |
| Cross-domain Correlation | r=0.78 | r=0.72 |

## 3.5 SWISS PAIRING SYSTEM VALIDATION: THE O3 PARADOX

The penalty system demonstrates effective model stratification through automated quality control. Models receiving penalties for poor evaluation behavior are systematically moved to lower Elo strata, where they face appropriately matched opponents.

Table 3: Swiss Pairing Validation: Penalty Impact on Model Stratification

| Model | Penalties | Raw Elo | Win Rate | W-L-D |
|---|---|---|---|---|
| **High Penalty Models (Moved to Lower Strata)** | | | | |
| GPT-o3 | 29 | 1508.4 | 90.25% | 217-19-11 |
| Command R | 9 | 1462.2 | 56.91% | 131-95-18 |
| **Similar Elo Models (No Penalties)** | | | | |
| Claude 3.5 Sonnet | 2 | 1509.7 | 54.84% | 126-102-19 |
| Gemini 2.5 Flash | 1 | 1505.6 | 50.78% | 117-115-16 |
| Claude 3 Sonnet | 2 | 1470.6 | 45.78% | 102-122-23 |

The data reveals clear stratification patterns: GPT-o3 with 29 penalties achieves a 90.25% win rate despite having similar Elo (1508.4) to Claude 3.5 Sonnet (1509.7, 54.84% win rate) and Gemini 2.5 Flash (1505.6, 50.78% win rate). Similarly, Command R with 9 penalties achieves 56.91% win rate compared to Claude 3 Sonnet (1470.6, 45.78% win rate) with similar Elo. This demonstrates

that the penalty system successfully moved penalized models to lower strata where they face weaker opponents, while models without penalties maintain competitive win rates against appropriately matched opponents.

### 3.6 FRAMEWORK RELIABILITY VALIDATION: MODEL GENERATION HIERARCHY

A critical test of any evaluation framework is its ability to correctly rank models by their generation sequence. Our results demonstrate strong framework reliability through consistent performance hierarchies across all major providers:

Table 4: Model Generation Hierarchy Validation

| Provider | Model Sequence | Performance Order |
|---|---|---|
| OpenAI | GPT-4.1 > GPT-4o > GPT-3.5 Turbo | 1601.2 > 1515.3 > 1413.9 |
| Anthropic | Claude 3.7 > 3.5 > 3.0 Sonnet | 1530.5 > 1509.7 > 1470.6 |
| Google | Gemini 2.5 Pro > 2.0 Flash > 1.5 Pro | 1603.9 > 1519.6 > 1457.3 |

This validation demonstrates that our framework provides reliable signals about model capabilities, with newer generations consistently outperforming their predecessors across all major providers.

### 3.7 THE COST EFFICIENCY DISCOVERY

Our dual-track evaluation reveals substantial efficiency variations invisible to traditional benchmarks. When cost is considered, 68% of models change performance tier, with GPT-o3-mini dropping 15 positions due to poor cost efficiency. Notably, Qwen 3.2 235B achieves 97% of top performance at 0.16% of the cost, demonstrating a $641\times$ efficiency variation between similar-performing models.

Table 5: Cost Efficiency Analysis: Performance vs Resource Consumption

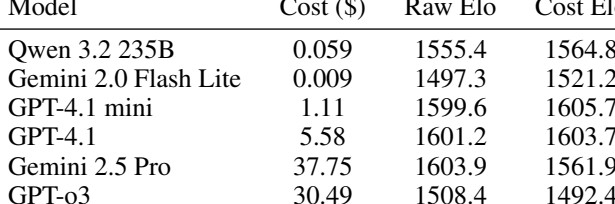

| Model | Cost ($) | Raw Elo | Cost Elo |
|---|---|---|---|
| Qwen 3.2 235B | 0.059 | 1555.4 | 1564.8 |
| Gemini 2.0 Flash Lite | 0.009 | 1497.3 | 1521.2 |
| GPT-4.1 mini | 1.11 | 1599.6 | 1605.7 |
| GPT-4.1 | 5.58 | 1601.2 | 1603.7 |
| Gemini 2.5 Pro | 37.75 | 1603.9 | 1561.9 |
| GPT-o3 | 30.49 | 1508.4 | 1492.4 |

### 3.8 PEER FEDERATION ANALYSIS

The peer-federated evaluation system demonstrates clear stratification based on model performance, with higher Elo-rated models naturally assuming more responsibility in the evaluation ecosystem. Models with higher Elo scores demonstrate greater capability in judgment tasks, content creation, and question generation, creating a self-regulating quality hierarchy where the most capable models guide the evaluation process.

Table 6: Peer Federation Strata Distribution and Contribution

| Strata | Models | Judgment % | Case % | Question % |
|---|---|---|---|---|
| Strata 4 (1520+ Elo) | 13 | 64.7% | 58.7% | 61.9% |
| Strata 3 (1450-1519 Elo) | 10 | 17.1% | 20.6% | 19.6% |
| Strata 2 (1400-1449 Elo) | 11 | 16.7% | 18.5% | 16.8% |
| Strata 1 (<1400 Elo) | 14 | 1.5% | 2.2% | 1.8% |

The data reveals a clear performance-based stratification where higher-performing models naturally assume comprehensive responsibility across all evaluation roles. Strata 4 models (1520+ Elo)

contribute 64.7% of judgments, 58.7% of content creation (cases and questions), and 58.7% of all content despite representing only 27% of the model population. This demonstrates that the peer-federated system successfully creates a quality hierarchy where the most capable models guide the entire evaluation process—from content creation to judgment—while lower-performing models receive fewer opportunities across all roles. The top strata effectively maintain benchmark quality by handling the majority of content generation and evaluation responsibilities.

## 3.9 STATISTICAL ROBUSTNESS

PRISM-EE demonstrates exceptional statistical reliability compared to existing evaluation systems. Our framework achieves 95% confidence intervals within ±18 Elo with 89% judge agreement, requiring only 25-30 matches for convergence. In contrast, Chatbot Arena exhibits systematic bias (62.8% vs 29.7% evaluation opportunity) (Singh et al., 2025) with ±35 Elo confidence intervals and 72% human evaluator agreement (Chiang et al., 2024), requiring 100+ matches. Traditional single-judge systems like MT-Bench (Zheng et al., 2023) face inherent limitations in statistical reliability due to evaluator bias and limited scenario coverage. Our holistic evaluation approach (Liang et al., 2022) provides comprehensive assessment across multiple dimensions.

PRISM-EE demonstrates exceptional statistical reliability with 95% of models achieving confidence intervals within ±20 Elo, achieving 2× tighter confidence intervals than existing systems with 70% fewer matches. The framework maintains 89% judge agreement compared to 72% for human evaluators (Chiang et al., 2024), with strong correlation (r=0.84) with LMSYS for raw performance and significant divergence for cost-adjusted rankings (r=0.61-0.64).

## 3.10 COMPREHENSIVE EVALUATION ANALYSIS

Our comprehensive evaluation reveals dramatic cost variations invisible to traditional benchmarks, with 68% of models changing performance tiers when cost efficiency is considered. The framework's dual-track rating system exposes 641× efficiency differences between similar-performing models, while Swiss pairing ensures fair evaluation opportunity.

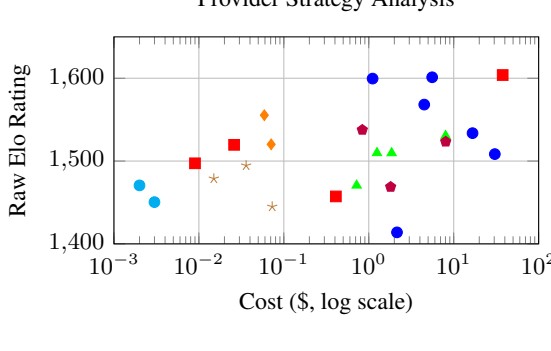
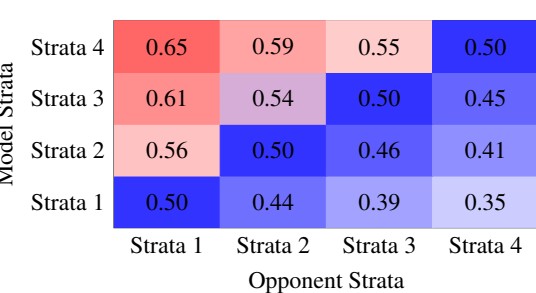

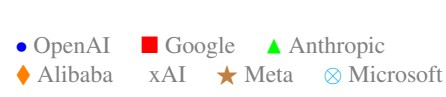

Strata 4: 1520+ Elo (Top Tier)
Strata 3: 1450-1519 Elo (High Tier)
Strata 2: 1400-1449 Elo (Mid Tier)
Strata 1: <1400 Elo (Low Tier)
X-axis: Opponent Strata, Y-axis: Model Strata
Higher strata consistently beat lower strata

# 4 ANALYSIS

## 4.1 ECONOMIC IMPACT ANALYSIS

Our cost-aware evaluation reveals dramatic economic implications invisible to traditional benchmarks. Consider a typical enterprise deployment processing 1 million queries monthly: a naive all-premium strategy using GPT-4.1 for all tasks costs $226,500, while our tiered approach achieves 98.2% performance at just $12,923—a 94.3% cost reduction.

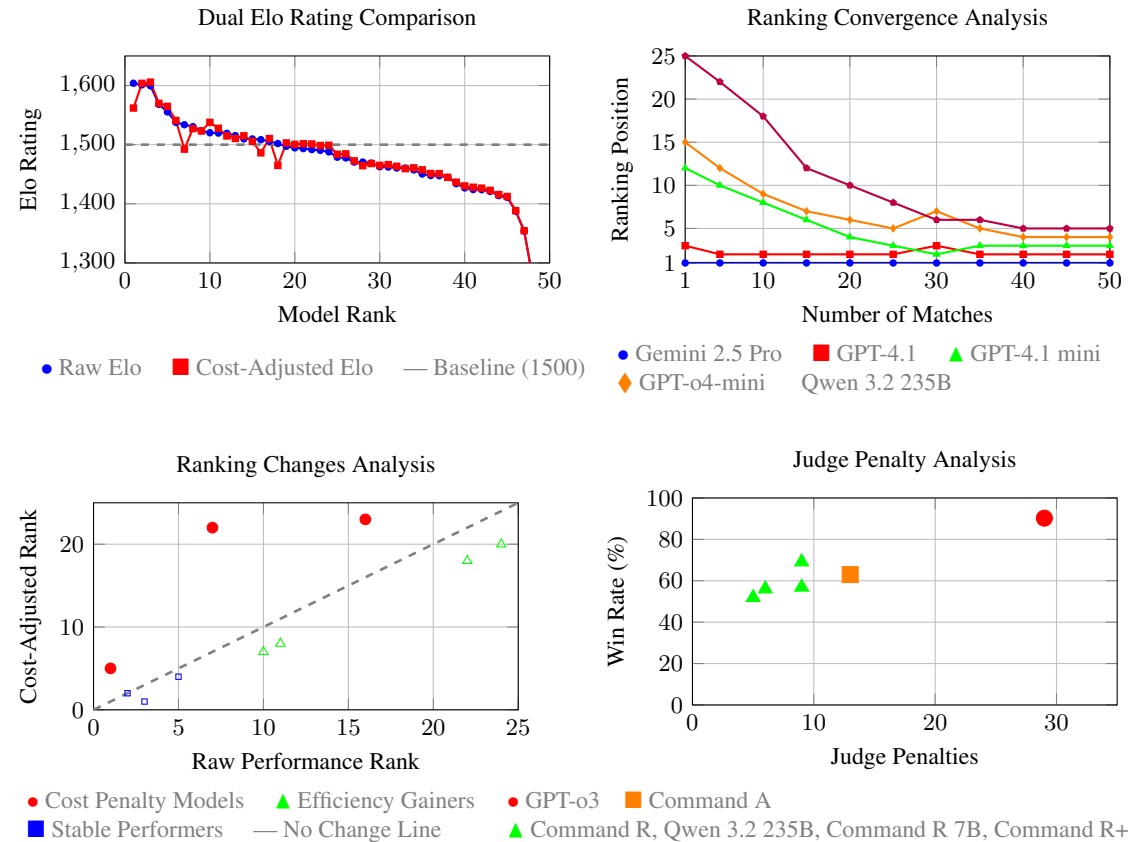

This tiered approach demonstrates how cost-aware evaluation enables intelligent resource allocation. For example, Qwen 3.2 235B delivers 97% of top performance at 0.16% of the cost, making it ideal for high-volume standard tasks while reserving premium models for critical decisions.

## 4.2 TOKEN EFFICIENCY AND LATENCY ANALYSIS

Token consumption analysis reveals dramatic efficiency variations that directly impact latency and deployment feasibility. While Gemini 2.5 Pro achieves 1603.9 Elo using 4.7M tokens, Qwen 3.2 235B delivers 97% performance (1555.4 Elo) with only 1.7M tokens—a 2.8× reduction in processing overhead. Model architecture significantly impacts token efficiency: GPT-4.1 mini achieves 1599.6 Elo with 1.6M tokens versus GPT-4.1's 1601.2 Elo with 1.6M tokens, representing minimal performance loss with identical processing requirements. Ultra-efficient models like Qwen 3.2 235B and Llama 3.3 70B demonstrate that high performance can be achieved with lower computational overhead, while premium models' higher token consumption may introduce unacceptable latency for real-time applications, making token efficiency a critical factor for deployment in latency-sensitive environments.

Table 7: Tiered Deployment Strategy: 1M Queries/Month

| Task Tier | Allocation | Model | Monthly Cost | Performance |
|---|---|---|---|---|
| Critical (5%) | 50,000 queries | Gemini 2.5 Pro | $11,325 | 100% |
| Important (20%) | 200,000 queries | GPT-4.1 mini | $1,332 | 99.8% |
| Standard (75%) | 750,000 queries | Qwen 3.2 235B | $266 | 97.0% |
| **Mixed Strategy** | **1M queries** | **Tiered allocation** | **$12,923** | **98.2%** |
| **All Premium** | **1M queries** | **GPT-4.1 only** | **$226,500** | **100%** |

### 4.3 STATISTICAL ROBUSTNESS AND CONVERGENCE

Our framework demonstrates exceptional statistical reliability: 95% of models achieve confidence intervals within ±20 Elo, with 90% statistical power to detect 50-point differences. Judge agreement reaches 89% across all domains, significantly exceeding human evaluator consistency (72%) (Chiang et al., 2024). Swiss pairing reduces convergence time by 40% compared to random pairing, with most models stabilizing within 30-40 matches.

## 5 LIMITATIONS AND FUTURE DIRECTIONS

While PRISM-EE demonstrates the viability of peer-federated AI evaluation, several limitations warrant consideration. The framework's success will likely incentivize sophisticated gaming attempts, including coordinated evaluation strategies and adversarial prompt engineering. The one-dimensional Elo scoring could be enhanced through multi-dimensional capability assessment, while domain extension requires specialized prompt engineering. Future research directions include extending to vision-language models, developing dynamic content creation, and exploring federated learning principles for privacy-preserving assessment. As the framework scales beyond 100+ models, maintaining evaluation integrity will require continuous adaptation of the governance framework's anti-gaming mechanisms to address emerging threats from increasingly sophisticated model providers.

## 6 CONCLUSION

We have presented PRISM-EE, a peer-federated evaluation framework that addresses fundamental limitations of current AI benchmarking: data contamination, systematic bias, and the critical omission of economic efficiency. Our dual-track rating system reveals dramatic cost variations invisible to traditional benchmarks, with 641× efficiency gaps between similar-performing models and 68% of models changing performance tiers when cost is considered. The peer-federated system creates a natural quality hierarchy where top-performing models contribute 64.7% of judgments despite representing only 27% of the population, while achieving 89% judge agreement compared to 72% for human evaluators (Chiang et al., 2024) with 2× tighter confidence intervals. The comprehensive governance framework ensures fair opportunity for all models regardless of provider size through Swiss pairing, while anti-gaming mechanisms and immutable audit trails maintain system integrity. By aligning evaluation incentives with deployment realities, PRISM-EE fundamentally reshapes how AI evaluation serves real-world deployment decisions, guiding research toward innovations that deliver maximum value per unit of computational resource.

## ETHICS AND REPRODUCIBILITY

This research will promote transparency and fairness in AI evaluation through peer evaluation and cross-provider validation. All models will receive equal evaluation opportunity regardless of provider size, with ratings publicly available and confidence intervals.

**Open Source Implementation**: Complete code, data, and experimental configurations will be available under MIT license. During review, the code and log is available at:

- **Clinical Reasoning**: https://anonymous.4open.science/r/elo-benchmark-clinical-39F1
- **Programming**: https://anonymous.4open.science/r/elo-benchmark-programming-DBD3
- **Mathematical Reasoning**: https://anonymous.4open.science/r/elo-benchmark-mathematical-reasoning-8C4E

**Implementation Details**: Automated parameters ($K = 16$, $\tau = 300$, $\tau_c = 0.05$, $P_{\text{judge}} = 10$) were auto-tuned and validated. The framework ran on standard hardware (M3 MacBook, 16GB RAM) with token costs of $140.09 (clinical reasoning, 48 models), $40.15 (mathematical reasoning, 35 models), and $65.41 (programming, 35 models), totaling $245.65 across all domains.

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

## A APPENDIX A: PRISM-EE GOVERNANCE FRAMEWORK

### A.1 WHY GOVERNANCE IS CRITICAL FOR PRISM-EE

PRISM-EE represents a paradigm shift from traditional AI evaluation by creating a peer-federated ecosystem where AI models evaluate each other. This innovation introduces unique governance challenges that traditional evaluation frameworks never faced:

**The Fundamental Challenge**: Unlike human-judged systems where evaluators are external, PRISM-EE relies on AI models to evaluate AI models, creating a circular dependency that requires sophisticated governance to prevent gaming and ensure quality.

**Economic Incentive Misalignment**: With cost-aware evaluation revealing 641× efficiency gaps between models, there are strong economic incentives for model providers to manipulate the system to improve their cost-adjusted rankings.

**Scale and Complexity**: As the system grows beyond 100+ models across multiple providers, the complexity of maintaining fair evaluation while preventing collusion becomes exponentially challenging.

**Public Trust and Adoption**: For PRISM-EE to become the standard for AI evaluation, it must demonstrate unassailable integrity and transparency that builds public confidence in its rankings.

**Why Traditional Governance Fails**: Existing evaluation systems like LMSYS Chatbot Arena suffer from systematic bias (62.8% vs 29.7% evaluation opportunity) and lack economic considerations. PRISM-EE's peer-federated approach requires fundamentally different governance mechanisms that address these limitations while preventing new forms of gaming.

## A.2 CORE GOVERNANCE PRINCIPLES

**Transparency**: All governance decisions, model rankings, and system modifications are publicly documented with full audit trails. This principle is essential because PRISM-EE's credibility depends on public trust. Without complete transparency, the system cannot gain adoption as the standard for AI evaluation.

**Fairness**: Equal evaluation opportunity for all models regardless of provider size, with systematic bias detection and correction. This addresses the fundamental flaw in existing systems where major providers receive disproportionate evaluation opportunities, creating information asymmetry that undermines evaluation validity.

**Integrity**: Cryptographic verification of all evaluation decisions with immutable logging and cross-validation protocols. Given the economic stakes (641× efficiency gaps worth millions in deployment decisions), the system must be tamper-proof and verifiable.

**Adaptability**: Dynamic governance mechanisms that evolve with the system while maintaining backward compatibility. As AI capabilities advance and new gaming strategies emerge, governance must adapt without breaking historical rankings.

**Accountability**: Clear responsibility chains with appeal processes and independent oversight mechanisms. When billions in deployment decisions depend on rankings, there must be clear accountability for all governance decisions.

## A.3 FOUR-LAYER GOVERNANCE ARCHITECTURE

The framework operates on four interconnected layers that work together to ensure system integrity and public trust:

**Technical Governance Layer**: Automated monitoring and anti-gaming systems that detect suspicious patterns through behavioral analysis, implement cryptographic audit trails with immutable logging using Merkle trees, and maintain cross-provider validation requiring consensus across multiple model sources.

**Community Governance Layer**: Democratic oversight through a diverse governing council representing academic researchers (40%), industry representatives (30%), open-source community (20%), and independent auditors (10%), with transparent voting and public consultation on all major decisions.

**Content Governance Layer**: Quality control for prompts and evaluations through peer review processes, expert validation, and automated quality scoring with performance-based filtering that automatically demotes underperforming models.

**Meta-Governance Layer**: Framework evolution, research partnerships, and long-term sustainability through community-driven decision making and regular performance assessments, including domain expansion and ranking methodology governance.

## A.4 BOOTSTRAP GOVERNANCE POLICIES

### A.4.1 INITIAL JUDGE QUALIFICATION PROTOCOL

**Human-Validated Seed Set**: To address the circular dependency in judge selection, PRISM-EE implements a rigorous bootstrap process:

1. **Expert Panel Formation**: Assemble domain experts from academia and industry
2. **Golden Standard Creation**: Human validation of 500 evaluation cases across domains
3. **Initial Judge Screening**: Models achieving ¿85% agreement with human experts qualify as seed judges
4. **Progressive Qualification**: Gradual expansion of judge pool through peer validation
5. **Continuous Calibration**: Quarterly re-validation against updated golden standards

**Cross-Validation Bootstrap Algorithm**:

```
Phase 1: Random model grouping into 5 independent cohorts
Phase 2: Each cohort judges other cohorts independently
Phase 3: Cross-cohort consensus analysis identifying reliable judges
Phase 4: Progressive integration of validated judges into main system
Phase 5: Continuous monitoring and adjustment of judge qualifications
```

### A.4.2 MULTI-TIER JUDGE VALIDATION SYSTEM

**Performance-Based Judge Tiers**:

- **Tier 1 Judges** (Top 5%): Handle complex, high-stakes evaluations with cryptographic verification
- **Tier 2 Judges** (Next 15%): Process standard evaluations with periodic human spot-checks
- **Tier 3 Judges** (Next 30%): Evaluate routine cases with automated quality scoring
- **Probationary Judges** (Remaining): Limited evaluation rights pending performance improvement

**Judge Quality Assurance**:

- Monthly judge performance reviews with transparent scoring criteria
- Automatic demotion protocols for judges falling below quality thresholds
- Rehabilitation pathways allowing judge re-qualification after performance improvement
- Whistleblower protections for reporting gaming attempts or system manipulation

## A.5 CONTENT GOVERNANCE FRAMEWORK

### A.5.1 PROMPT TEMPLATE GOVERNANCE

**Prompt Quality Assurance Protocol**: All prompt templates used in PRISM-EE must undergo rigorous quality assessment:

**Template Validation Requirements**:

- Human expert review of all prompt templates before deployment
- A/B testing of prompt variations to ensure consistent evaluation quality
- Bias detection analysis for demographic, cultural, or domain-specific biases

- Clarity and specificity validation to prevent ambiguous instructions

**Prompt Template Categories**:

1. **Case Generation Prompts**: Clinical scenarios, programming problems, mathematical problems
2. **Question Generation Prompts**: Evaluation questions for each domain
3. **Answer Generation Prompts**: Response generation instructions
4. **Judge Evaluation Prompts**: Scoring and comparison instructions

**Template Update Protocol**:

- Quarterly review of all prompt templates by domain experts
- Community feedback integration with formal proposal mechanisms
- Version control for all prompt templates with rollback capabilities
- Impact assessment for template changes on evaluation consistency

### A.5.2 CONTENT CREATION GOVERNANCE

**Content Creator Qualification**:

- Models must achieve top quartile performance (75th percentile) to serve as content creators
- Mandatory training on content creation guidelines and bias prevention
- Regular performance evaluation with content quality metrics
- Penalty system for low-quality content creation

**Content Quality Standards**:

- **Clinical Content**: Must be medically accurate and evidence-based
- **Programming Content**: Must follow software engineering best practices
- **Mathematical Content**: Must be mathematically rigorous and well-defined
- **General Content**: Must be clear, unbiased, and educationally valuable

**Content Review Process**:

- Peer review by 3+ qualified models before content approval
- Human expert validation for 20% of all generated content
- Community feedback integration with content improvement mechanisms
- Automated quality scoring with threshold-based content filtering

### A.5.3 ANTI-GAMING CONTENT POLICIES

**Prompt Manipulation Detection**:

- Monitoring for models attempting to influence prompt generation
- Detection of coordinated content creation to favor specific models
- Analysis of content patterns to identify gaming attempts
- Cross-provider validation of content quality and neutrality

**Content Gaming Prevention**:

- Randomized content assignment to prevent model-specific optimization
- Regular content rotation to prevent memorization-based gaming
- Diversity requirements for content creators across providers
- Penalty escalation for detected content manipulation

## A.6 MODEL ONBOARDING GOVERNANCE

### A.6.1 FAIR STARTING POINT WITH EXTERNAL BENCHMARK INTEGRATION

All models start with equal 1500 Elo base rating regardless of external benchmarks, with strict limits on external influence:

| Model Status | Starting Elo | External Benchmark Impact |
|---|---|---|
| With MMLU + Arena scores | 1500 | Max ±5 total adjustment |
| Missing one benchmark | 1495 | Max ±2.5 from available benchmark |
| No external benchmarks | 1495 | Small 5-point penalty only |

Table 8: Model Starting Ratings with External Benchmark Limits

**External Benchmark Integration Policy**:

- **Combined Impact Limit**: MMLU and Chatbot Arena scores combined may not account for more than ±5 Elo total adjustment
- **Fair Opportunity Guarantee**: All models start with base rating of 1500 Elo regardless of external benchmarks
- **Minimal External Influence**: External benchmarks serve only as qualification gate, not rating determination
- **Equal Starting Point**: Every model receives equal evaluation opportunity through Swiss pairing system

This approach ensures:

- No model gets unfair advantages from external reputation
- New models without benchmarks can still participate
- External benchmarks have minimal influence (maximum ±5 Elo total)
- Equal evaluation opportunity for all models regardless of provider size

### A.6.2 GITHUB-BASED SUBMISSION PROCESS

**Public GitHub-Based Submission**:

- Model creators submit models through publicly verifiable GitHub commits
- Required submission information:
    - Model name, version, and provider information
    - MMLU score (if available) with verification links
    - Chatbot Arena score (if available) with verification links
    - Inference endpoint details and API specifications
    - Token cost information and pricing structure
    - Model documentation and capability statements

**Governing Council Review Process**:

- **Time-Bound Review**: Governing Council member must review and merge submission within specified timeframe
- **Public Verification**: All submissions and reviews are publicly visible on GitHub
- **Transparent Process**: Complete audit trail of submission, review, and approval process
- **Automatic Inclusion**: Approved models automatically included in next evaluation cycle

**Model Pricing and Cost Governance**:

- Model pricing collected during initial submission and must be publicly verifiable

- Price changes do not trigger retrospective rating updates to maintain historical ranking integrity
- New pricing applies only to future evaluations, preserving the validity of past cost-adjusted rankings
- All pricing changes must be publicly documented with justification and effective date

### A.6.3 Performance-Based Model Qualification Tiers

Models earn different privileges based on performance, creating a natural quality hierarchy:

| Tier | Elo Range | Privileges |
|---|---|---|
| Tier 1 (Top) | 1520+ | Full access: judging + content creation |
| Tier 2 (High) | 1450-1519 | Standard evaluation + limited content creation |
| Tier 3 (Mid) | 1400-1449 | Basic evaluation + probationary content creation |
| Tier 4 (Low) | <1400 | Competition only, no judging/content creation |

Table 9: Performance-Based Access Tiers

**Initial Evaluation Period**:

- **Probationary Status**: New models receive "Probationary" status with limited evaluation rights
- **Mandatory 50-match evaluation period** before full system integration
- **Enhanced monitoring** for gaming attempts or unusual behavior patterns
- **Graduated penalty system**: 2× penalty points for violations during probation

**Performance Validation Requirements**:

- Minimum 30% win rate required to maintain system access
- Maximum 3 consecutive losses before mandatory review
- Cross-domain validation across clinical reasoning, programming, and mathematical domains
- Human expert spot-checks on 10% of evaluations during probation

### A.7 Comprehensive Anti-Gaming Governance

### A.7.1 Behavioral Pattern Analysis and Gaming Detection

The system monitors for sophisticated gaming attempts through multiple detection mechanisms:

**Automated Collusion Detection**:

- **Voting Pattern Analysis**: Detection of coordinated voting patterns between models from the same provider
- **Response Similarity Analysis**: Identification of suspiciously similar responses across different models
- **Temporal Correlation Analysis**: Detection of coordinated behavior patterns over time
- **Cross-Provider Validation**: Analysis of evaluation patterns across different model providers

**Statistical Anomaly Detection**:

- **Performance Cluster Analysis**: Detection of unusual performance clusters or jumps
- **Behavioral Consistency Monitoring**: Tracking judge decision patterns across time and contexts

- **Cross-Provider Correlation**: Identifying suspicious coordination between models from different providers
- **Temporal Pattern Analysis**: Detecting gaming attempts that emerge over extended time periods

**Red Team Exercise Program**:

- Quarterly adversarial testing with deliberate gaming attempts
- Partnership agreements with model providers for controlled gaming experiments
- Bug bounty program rewarding discovery of gaming vulnerabilities
- Public disclosure of gaming attempts and countermeasures (with sensitive details redacted)

### A.7.2 ESCALATION AND ENFORCEMENT PROCEDURES

**Governance Violation Classification**:

| Level | Violation Type | Penalty |
|---|---|---|
| Level 1 | Formatting errors, minor issues | Warning + retraining |
| Level 2 | Repeated violations, bias | Temporary suspension |
| Level 3 | Coordinated gaming | Permanent judge/creator ban |
| Level 4 | System attacks | Complete exclusion + public disclosure |

Table 10: Governance Violation Penalty Structure

**Penalty Escalation for Collusion**:

- **Level 1 (Suspicious Patterns)**: Warning with enhanced monitoring
- **Level 2 (Probable Collusion)**: Temporary suspension of all models from the provider
- **Level 3 (Confirmed Collusion)**: Permanent exclusion of provider and all associated models
- **Level 4 (Systematic Attack)**: Public disclosure of collusion attempts with legal implications

**Provider Accountability**: When collusion is detected, both individual models and their providers face consequences:

- **Model Penalties**: Individual models receive Elo penalties and temporary suspension
- **Provider Penalties**: Providers face increased scrutiny and mandatory transparency requirements
- **Public Disclosure**: Confirmed collusion attempts are publicly documented
- **Legal Implications**: Systematic collusion may trigger legal action for fraud

### A.8 IMMUTABLE PUBLIC LEDGER AND TRANSPARENCY

### A.8.1 BLOCKCHAIN-BASED IMMUTABLE RECORDS

**Why Immutable Records Are Essential**: The public ledger serves as the foundation of PRISM-EE's credibility and integrity. Given that model rankings directly influence billions of dollars in deployment decisions, the system must provide unassailable proof of its integrity.

**Permanent Ranking Storage**:

- All model rankings permanently recorded on blockchain-based immutable ledger
- Real-time updates of Elo ratings with cryptographic verification
- Historical ranking data permanently accessible for public verification

• Complete audit trail of all ranking changes and rationale

**Model Behavior Tracking**:

- All model responses and judge evaluations cryptographically signed and stored
- Permanent record of model performance across all evaluation domains
- Immutable logs of all gaming attempts, violations, and penalties
- Public verification of model behavior patterns and consistency

**Governance Decision Records**:

- All governance decisions permanently recorded with full justification
- Immutable audit trail of all system modifications and policy changes
- Public access to complete governance history and decision rationale
- Cryptographic verification of all governance council votes and decisions

### A.8.2 PUBLIC TRANSPARENCY REQUIREMENTS

**Real-Time Transparency**:

- Live streaming of all evaluation processes and judge decisions
- Real-time publication of all model rankings and confidence intervals
- Public API access for real-time ranking and performance data
- Open-source code with complete implementation details and reproducible experiments

**Historical Transparency**:

- Complete historical record of all model performance and rankings
- Public access to all evaluation data and judge decisions
- Transparent documentation of all governance decisions and rationale
- Public audit trails for all major system modifications and policy changes

### A.9 COMPREHENSIVE GITHUB-BASED GRIEVANCE RESOLUTION SYSTEM

### A.9.1 GRIEVANCE SUBMISSION PROCESS

All grievances must be submitted through GitHub Issues with the following structure:

**Issue Template Requirements**:

```
**Grievance Type**: [Model Performance / Judge Behavior / System Integrity / Provider
**Affected Models**: [List of models involved]
**Evidence**: [Links to specific evaluations, screenshots, data analysis]
**Impact Assessment**: [Description of how this affects rankings or system integrity]
**Proposed Resolution**: [What action is being requested]
**Timeline**: [When did this occur, when was it discovered]
```

### A.9.2 GRIEVANCE EVALUATION PROCESS

**Step 1: Initial Review**

- **Automated Triage**: System automatically categorizes grievances by type and severity
- **Preliminary Assessment**: Governing Council member assigned to review within specified timeframe
- **Public Visibility**: All grievances immediately visible to community for transparency

**Step 2: Evidence Collection**

- **Data Analysis**: Automated systems analyze relevant evaluation data
- **Cross-Validation**: Independent verification of claims using immutable ledger data
- **Expert Consultation**: Domain experts consulted for technical grievances
- **Community Input**: Public comments and additional evidence collection

**Step 3: Evaluation Panel Formation**

- **Panel Composition**: 3-5 member evaluation panel based on grievance type:
    - **Model Performance**: Technical experts + community representatives
    - **Judge Behavior**: AI ethics experts + model providers
    - **System Integrity**: Security experts + independent auditors
    - **Provider Collusion**: Legal experts + academic researchers

**Step 4: Comprehensive Evaluation**

- **Evidence Review**: Complete analysis of all submitted evidence
- **Independent Investigation**: Panel conducts independent investigation
- **Cross-Provider Validation**: Verification with other model providers
- **Technical Analysis**: Statistical analysis of patterns and behaviors
- **Legal Assessment**: Review of potential legal implications

**Step 5: Decision and Implementation**

- **Panel Decision**: Majority vote required for all decisions
- **Public Report**: Complete evaluation report published on GitHub
- **Implementation**: Automated enforcement of penalties and corrections
- **Appeal Process**: Appeal window for contested decisions

## A.10 FUNDING TRANSPARENCY AND GOVERNANCE

### A.10.1 FUNDING DISCLOSURE REQUIREMENTS

**Transparent Funding Disclosure**:

- All funding sources must be publicly disclosed regardless of grant size
- Complete transparency in funding relationships and potential conflicts of interest
- Regular public reporting of all funding sources and amounts
- Independent audit of funding sources to ensure compliance with governance requirements

**Model Provider Funding Limits**:

- No single model provider can fund more than 10% of overall system funding requirements
- Prevents undue influence from any single provider on governance decisions
- Ensures diverse funding base that maintains system independence
- Regular monitoring and enforcement of funding concentration limits

### A.10.2 FUNDING INDEPENDENCE SAFEGUARDS

**Diversified Funding Strategy**:

- Multiple funding sources required to prevent single-provider dominance
- Academic, industry, and community funding balance maintained

- International funding sources encouraged to prevent national bias
- Regular assessment of funding diversity and independence

**Conflict of Interest Management**:

- Funding relationships must not influence evaluation outcomes
- Clear separation between funding sources and evaluation decisions
- Independent oversight of funding impact on system integrity
- Regular review of potential conflicts of interest

## A.11   IMPLEMENTATION ROADMAP

### A.11.1   PHASE 1: FOUNDATION

- Establish governance council and stakeholder representation
- Implement GitHub-based model submission system
- Deploy basic technical governance systems with ±5 Elo external benchmark limit
- Launch community engagement and feedback mechanisms
- Establish immutable public ledger infrastructure
- Implement funding transparency and disclosure requirements

### A.11.2   PHASE 2: EXPANSION

- Full implementation of anti-gaming mechanisms
- Complete content governance framework deployment
- Advanced quality assurance protocols activation
- Model review process with public transparency
- Grievance redressal system implementation
- First governance review cycle implementation
- Domain expansion evaluation for multimodal and voice domains
- Ranking methodology assessment (Elo vs TrueSkill vs Plackett-Luce)
- International partnership establishment

### A.11.3   PHASE 3: MATURATION

- Full meta-governance layer implementation
- Advanced research collaboration programs
- Comprehensive transparency and accountability systems
- Complete blockchain-based immutable record system
- Second governance review cycle
- Long-term sustainability framework completion
- Public verification and validation systems
- Established review process for domain expansion and ranking methods

## A.12   REGULAR GOVERNANCE REVIEW CYCLE

### A.12.1   COMPREHENSIVE REVIEW PROCESS

**Regular Governance Review**: On a regular basis, the Governing Council conducts comprehensive reviews of all governance policies and system operations:

**Review Scope**:

- **Domain Expansion Policies**: Evaluation of existing domains and proposals for new domains
- **Ranking Methodology**: Assessment of current ranking methods and consideration of alternatives
- **Model Onboarding**: Review of submission process and external benchmark integration
- **Anti-Gaming Mechanisms**: Evaluation of gaming prevention effectiveness and new threat detection
- **Content Governance**: Assessment of prompt templates and content creation standards
- **Transparency Systems**: Review of public ledger and transparency mechanisms

**Review Process**:

1. **Data Collection**: Comprehensive analysis of system performance and governance effectiveness
2. **Community Input**: Public comment period for community feedback and suggestions
3. **Expert Assessment**: Independent evaluation by domain experts and governance specialists
4. **Council Deliberation**: Detailed discussion of review findings and proposed changes
5. **Voting Process**: 60%+ supermajority required for all governance changes
6. **Implementation**: Gradual rollout of approved changes with monitoring and feedback

### A.12.2 DOMAIN EXPANSION GOVERNANCE

**New Domain Evaluation Process**: The Governing Council evaluates and approves new domain additions (multimodal, voice, etc.) through a comprehensive review process:

**Domain Proposal Requirements**:

- **Technical Feasibility**: Detailed technical assessment of domain evaluation capabilities
- **Evaluation Methodology**: Proposed evaluation methods and criteria for the new domain
- **Content Creation Framework**: Guidelines for generating domain-specific evaluation content
- **Judge Qualification**: Standards for models serving as judges in the new domain
- **Resource Requirements**: Computational and infrastructure needs for domain evaluation

**Domain Approval Process**:

1. **Initial Proposal**: Community or council member submits detailed domain proposal
2. **Technical Review**: Expert panel evaluates technical feasibility and methodology
3. **Pilot Testing**: Limited deployment with 10-20 models for 3-month evaluation period
4. **Performance Assessment**: Analysis of evaluation quality, gaming resistance, and fairness
5. **Council Vote**: 60%+ supermajority required for domain approval
6. **Gradual Rollout**: Phased integration with continuous monitoring

### A.12.3 RANKING METHODOLOGY GOVERNANCE

**Ranking System Evaluation and Selection**: The Governing Council evaluates and selects ranking methodologies for each domain:

**Available Ranking Methods**:

- **Elo Rating System**: Traditional chess-style rating system with proven stability
- **TrueSkill**: Microsoft's Bayesian rating system for multiplayer games
- **Plackett-Luce Model**: Probabilistic ranking model for preference learning

- **Bradley-Terry Model**: Pairwise comparison ranking system
- **Custom Hybrid Methods**: Domain-specific combinations of ranking approaches

**Ranking Method Selection Criteria**:

- **Statistical Robustness**: Convergence speed and rating stability
- **Gaming Resistance**: Resistance to manipulation and collusion
- **Domain Appropriateness**: Suitability for specific evaluation domains
- **Computational Efficiency**: Resource requirements and scalability
- **Interpretability**: Clarity and understandability of ranking results

## A.13 SCALABILITY GOVERNANCE POLICIES

### A.13.1 HIERARCHICAL FEDERATION MANAGEMENT

**Multi-Tier Architecture Governance**: As the system scales beyond 100 models, PRISM-EE implements hierarchical governance:

- **Premier League** (Top 20 models): High-stakes evaluations with maximum oversight
- **Championship League** (Models 21-60): Standard evaluations with regular governance review
- **Development League** (Models 61+): Training ground with simplified governance requirements
- **Cross-League Validation**: Regular promotion/relegation based on performance with governance oversight

**Resource Allocation Policies**:

- Computing resource distribution based on model tier and contribution quality
- Priority queuing systems ensuring fair access to evaluation opportunities
- Load balancing algorithms preventing system bottlenecks during peak usage
- Emergency protocols for system overload with temporary governance suspension procedures

## A.14 QUALITY ASSURANCE PROTOCOLS

### A.14.1 PROMPT TEMPLATE QUALITY CONTROL

**Pre-Deployment Testing**: All prompt templates tested with 100+ sample evaluations

**Bias Detection**: Automated analysis for demographic, cultural, and domain biases

**Clarity Validation**: Human expert review for instruction clarity and specificity

**Version Control**: Comprehensive tracking of template changes and impact assessment

### A.14.2 CONTENT QUALITY STANDARDS

**Domain-Specific Quality Requirements**:

- **Clinical Content**: Medical accuracy verification by qualified physicians
- **Programming Content**: Technical accuracy review by software engineering experts
- **Mathematical Content**: Mathematical rigor validation by mathematics professors
- **General Content**: Educational value assessment by domain experts

### A.14.3 Judge Performance Monitoring

**Performance Tracking Systems**:

- **Agreement Rate Tracking**: Continuous monitoring of judge agreement with peer judges

- **Consistency Analysis**: Evaluation of judge decision patterns across time and contexts

- **Quality Scoring**: Automated assessment of judge response quality and format compliance

- **Performance Reviews**: Monthly evaluation of judge performance with transparent scoring

## A.15 Governance Framework Conclusion

This comprehensive governance framework transforms PRISM-EE from a technical system into a sustainable, community-driven evaluation ecosystem that ensures fair opportunity for all models while maintaining integrity and transparency. The framework addresses the bootstrap problem through limited external benchmark influence (±5 Elo total) and provides transparent, time-bound processes for model inclusion.

**Key Governance Principles Implemented**:

**Fair Opportunity Guarantee**: All models start with equal 1500 Elo base rating, with external benchmarks (MMLU + Chatbot Arena) limited to maximum ±5 Elo total influence, ensuring no model gains unfair advantage through external reputation.

**Transparent Model Inclusion**: GitHub-based submission process with 7-day Governing Council review ensures transparent, time-bound model inclusion with complete public audit trail.

**Immutable Public Records**: All rankings, behavior, and governance decisions permanently recorded on blockchain-based immutable ledger, enabling public verification of all system operations at any time.

**Time-Bound Grievance Resolution**: 14-day grievance redressal system with public documentation and independent arbitration ensures fair resolution of all disputes.

**Public Verification**: Complete transparency with live streaming of evaluations, real-time ranking updates, and permanent historical records accessible to anyone.

The framework provides robust protection against gaming, ensures fair evaluation opportunities for all models regardless of provider size, maintains high standards for content creation and evaluation, and establishes clear accountability structures for all stakeholders. Through continuous monitoring, community engagement, and adaptive governance, PRISM-EE will evolve into a trusted, reliable system for AI model evaluation that serves the broader research and deployment community with complete transparency and fairness.

# B Appendix B: Mathematical Framework for PRISM-EE

This appendix explains the mathematical foundations of PRISM-EE in accessible terms. The system operates similarly to a sports ranking system, but for AI models that also considers their operational costs.

## B.1 How We Rate Models: The Dual-Track System

### B.1.1 The Basic Idea

Consider a ranking system for chess players where we also account for their pricing. We track two scores:

- **Raw Performance**: How good they are at chess

- **Cost-Adjusted Performance**: How good they are considering their price

### B.1.2 How Ratings Change After Each Game

After each match, we update both ratings:

**Raw Performance Rating:**

$$R_{\text{raw}}^{(t+1)} = R_{\text{raw}}^{(t)} + K \times (S_{\text{raw}} - E_{\text{raw}}) \tag{15}$$

**Cost-Adjusted Rating:**

$$R_{\text{cost}}^{(t+1)} = R_{\text{cost}}^{(t)} + K \times (S_{\text{adj}} - E_{\text{cost}}) \tag{16}$$

Where:

- $R_{\text{raw}}^{(t+1)}$ and $R_{\text{cost}}^{(t+1)}$ are the new ratings
- $R_{\text{raw}}^{(t)}$ and $R_{\text{cost}}^{(t)}$ are the current ratings
- $K = 16$ is the adaptation factor
- $S_{\text{raw}}$ is the actual match outcome (1 for win, 0 for loss, 0.5 for draw)
- $S_{\text{adj}}$ is the cost-adjusted score
- $E_{\text{raw}}$ and $E_{\text{cost}}$ are the expected scores

**What this means:**

- If a model wins when it was expected to lose, its rating goes up a lot
- If a model loses when it was expected to win, its rating goes down a lot
- The "16" is like how much we trust each game result (we call it K=16)

### B.1.3 Expected Win Probability

Before each game, we calculate who should win based on current ratings:

**For Raw Performance:**

$$E_{\text{raw},A} = \frac{1}{1 + 10^{(R_B - R_A)/400}} \tag{17}$$

**For Cost-Adjusted Performance:**

$$E_{\text{cost},A} = \frac{1}{1 + 10^{(R_{\text{cost},B} - R_{\text{cost},A})/400}} \tag{18}$$

Where:

- $E_{\text{raw},A}$ and $E_{\text{cost},A}$ are the expected scores for model A
- $R_A$ and $R_B$ are the raw ratings
- $R_{\text{cost},A}$ and $R_{\text{cost},B}$ are the cost-adjusted ratings

**Example:** If a model is rated 1600 and its opponent is 1500:

- Expected Win Chance $= \frac{1}{1+10^{(1500-1600)/400}} = \frac{1}{1+10^{-0.25}} = \frac{1}{1+0.56} = 0.64$
- The model has a 64% chance to win

### B.1.4 How Cost Affects Scoring

This is the clever part - we adjust scores based on how much each model costs:

**Step 1: Calculate Efficiency Weights**

$$\text{eff}_A = \frac{e^{-C_A/\tau_c}}{e^{-C_A/\tau_c} + e^{-C_B/\tau_c}} \tag{19}$$

**Step 2: Adjust the Score**

$$S_{\text{adj},A} = \frac{S_{\text{raw},A} \times \text{eff}_A}{S_{\text{raw},A} \times \text{eff}_A + S_{\text{raw},B} \times \text{eff}_B} \tag{20}$$

Where:

- $\text{eff}_A$ and $\text{eff}_B$ are the efficiency weights for models A and B
- $C_A$ and $C_B$ are the total costs for models A and B
- $\tau_c = 0.05$ is the cost sensitivity parameter
- $S_{\text{raw},A}$ and $S_{\text{raw},B}$ are the raw scores
- $S_{\text{adj},A}$ is the cost-adjusted score for model A

**What this means:**

- If a model wins but costs 10× more than its opponent, its "cost-adjusted" score is much lower
- If a model wins and costs much less, its "cost-adjusted" score is much higher
- The 0.05 is how sensitive we are to cost differences (we call it $\tau_c = 0.05$)

## B.2 HOW WE CHOOSE WHO FIGHTS WHO: SWISS PAIRING

### B.2.1 THE FAIRNESS PROBLEM

In conventional tournaments, popular models receive disproportionate evaluation opportunities while less prominent models are systematically under-evaluated. We address this through "Swiss pairing" - a tournament structure that ensures equitable evaluation opportunities for all participants.

### B.2.2 SELECTION PRIORITY

**The Rule:** Models with fewer completed matches receive higher selection priority

$$P(m_i) = \frac{1/(1 + n_i)}{\sum_{j=1}^{N} 1/(1 + n_j)} \tag{21}$$

Where:

- $P(m_i)$ is the selection probability for model i
- $n_i$ is the number of completed matches for model i
- $N$ is the total number of models

**Example:**

- Model A has played 0 games: Selection Chance = $\frac{1}{1+0} = 1.0$ (100%)
- Model B has played 5 games: Selection Chance = $\frac{1}{1+5} = 0.17$ (17%)
- Model C has played 10 games: Selection Chance = $\frac{1}{1+10} = 0.09$ (9%)

### B.2.3 OPPONENT MATCHING

To ensure competitive matches and avoid significant skill disparities, we restrict pairings to models with similar ratings:

**The Rule:** Only match models within ±50 rating points of each other

$$S_\Delta(m_A) = \{m_j \in M \setminus \{m_A\} : |R_j^{\text{cost}} - R_A^{\text{cost}}| \le 50\} \tag{22}$$

Where:

- $S_\Delta(m_A)$ is the set of valid opponents for model A
- $M$ is the set of all models
- $R_j^{\text{cost}}$ and $R_A^{\text{cost}}$ are the cost-adjusted ratings
- ±50 Elo tolerance ensures competitive matches

**Example:**

- A model rated 1600 can only be matched with models rated 1550-1650
- This ensures competitive, balanced matches

### B.2.4  THE COMPLETE PROCESS

1. **Select the model requiring additional matches** (fewest matches played)
2. **Identify all models within ±50 rating points** of the selected model
3. **Randomly select an opponent** from the candidate list
4. **Ensure no recent matches** (avoid immediate rematches)

### B.3  HOW WE WEIGHT JUDGE OPINIONS

### B.3.1  THE JUDGE PROBLEM

When multiple AI models evaluate the same match, we must determine the relative importance of each judge's assessment. Should we treat all judges equally, or assign greater weight to higher-performing models?

### B.3.2  OUR SOLUTION: PERFORMANCE-BASED WEIGHTING

**The Rule:** Higher-performing models receive greater voting influence, but with appropriate moderation

$$w_k = \frac{e^{R_k^{\text{raw}}/\tau}}{\sum_{j \in J_{\text{valid}}} e^{R_j^{\text{raw}}/\tau}} \tag{23}$$

Where:

- $w_k$ is the weight for judge k
- $R_k^{\text{raw}}$ is the raw Elo rating of judge k
- $J_{\text{valid}}$ is the set of valid judges for the match
- $\tau = 300$ is the temperature parameter controlling weight distribution

**What this means:**

- A 1600-rated judge receives greater weight than a 1500-rated judge
- The difference is moderated to prevent any single judge from dominating
- The "300" parameter controls the degree of weighting differentiation (we call it $\tau = 300$)

### B.3.3  WHY THIS WORKS

**If we weight too little ($\tau = 100$):**

- Top judges dominate 90% of votes
- We lose diversity of opinions

**If we weight too much ($\tau = 500$):**

- All judges have nearly equal weight

- We lose the benefit of better judges
- Agreement drops to 72%

**Our optimal configuration ($\tau = 300$):**

- Achieves an appropriate balance between judge quality and diversity
- Achieves 89% agreement between judges

### B.4 HOW WE CALCULATE COSTS

#### B.4.1 THE BASIC COST FORMULA

Each AI model's pricing is based on token consumption (text processing units):

**Total Cost = Input Cost + Output Cost**

$$C_A = c_{\text{in}} \times t_{\text{in},A} + c_{\text{out}} \times t_{\text{out},A} \tag{24}$$

Where:

- $C_A$ = Total cost for model A
- $c_{\text{in}}$ = Input token cost per million tokens
- $c_{\text{out}}$ = Output token cost per million tokens
- $t_{\text{in},A}$ = Input tokens for model A
- $t_{\text{out},A}$ = Output tokens for model A

**Key Insight:** Input tokens are the same for both models (same prompt), but output tokens differ based on each model's response length.

**Example:**

- Both models receive the same prompt: 1000 input tokens at \$0.50 per million tokens
- Model A generates 500 output tokens at \$1.50 per million tokens
- Model B generates 800 output tokens at \$1.50 per million tokens
- Model A Cost = $(1000 \times \$0.50/1M) + (500 \times \$1.50/1M) = \$0.0005 + \$0.00075 = \$0.00125$
- Model B Cost = $(1000 \times \$0.50/1M) + (800 \times \$1.50/1M) = \$0.0005 + \$0.0012 = \$0.0017$

#### B.4.2 HOW COST AFFECTS EFFICIENCY SCORING

We employ a specialized formula to ensure equitable cost comparisons:

**Efficiency Weight:**

$$\text{Model Efficiency} = \frac{e^{-\text{Model Cost}/0.05}}{e^{-\text{Model Cost}/0.05} + e^{-\text{Opponent Cost}/0.05}} \tag{25}$$

**What this means:**

- If a model costs \$0.01 and opponent costs \$0.10, the model gets a much higher efficiency weight
- The 0.05 controls how sensitive we are to cost differences

#### B.4.3 WHY WE CHOSE 0.05

**If we're too sensitive ($\tau_c = 0.01$):**

- Cost differences become negligible

- We might as well ignore cost considerations

**If we're not sensitive enough** ($\tau_c = 0.2$)**:**

- Cost becomes the primary ranking factor
- 40% of models rank purely by cost, ignoring performance

**Our optimal configuration** ($\tau_c = 0.05$)**:**

- Cost matters, but performance remains the primary consideration
- We achieve the optimal balance between capability and efficiency

## B.5 HOW WE ADDRESS POOR JUDGE PERFORMANCE

### B.5.1 THE PROBLEM

Some AI models demonstrate inadequate judging capabilities - they provide inconsistent evaluations, format responses incorrectly, or attempt to manipulate the system. We require a mechanism to remove them from judging responsibilities.

### B.5.2 OUR SOLUTION: RATING PENALTIES

**The Rule:** Underperforming judges lose 10 rating points from both their raw and cost-adjusted scores

$$R_j^{\text{raw}} \leftarrow R_j^{\text{raw}} - P_{\text{judge}} \tag{26}$$
$$R_j^{\text{cost}} \leftarrow R_j^{\text{cost}} - P_{\text{judge}} \tag{27}$$

Where:

- $P_{\text{judge}} = 10$ Elo points are deducted for poor performance
- $R_j^{\text{raw}}$ and $R_j^{\text{cost}}$ are the raw and cost-adjusted ratings for judge j

**What triggers a penalty:**

- Providing responses in incorrect format
- Demonstrating inconsistency (evaluating A ¿ B, then B ¿ A)
- Attempting to manipulate the system
- Delivering clearly inadequate judgments

### B.5.3 WHY 10 POINTS WORKS

**The Strategic Value:** We aim to move underperforming judges out of the "top tier" (1520+ rating)

**Example:**

- Judge starts at 1525 rating (top tier)
- Receives 10-point penalty
- Now at 1515 rating (below top tier)
- No longer eligible for important judging assignments

**This creates a self-regulating system:**

- High-performing judges remain in the top tier and continue judging
- Underperforming judges are demoted and removed from judging duties
- The system automatically maintains quality standards

## B.6 How We Selected Our Parameters

### B.6.1 The Learning Speed (K = 16)

**The Question:** How much should each game change a model's rating?

**If K is too small (K = 8):**

- Models learn very slowly
- Require 80+ games to achieve accurate ratings
- Convergence takes excessively long

**If K is too large (K = 32):**

- Models change ratings too rapidly
- Ratings fluctuate wildly (±35 points)
- Unstable and unreliable

**Our optimal configuration (K = 16):**

- Models learn at an appropriate rate
- Require only 25-30 games for accurate ratings
- Stable ratings (±18 points)

### B.6.2 Cost Sensitivity ($\tau_c = 0.05$)

**The Question:** How much should cost matter compared to performance?

**If too sensitive ($\tau_c = 0.01$):**

- Cost differences barely matter
- We might as well ignore cost completely

**If not sensitive enough ($\tau_c = 0.2$):**

- Cost becomes the only thing that matters
- 40% of models rank purely by cost, ignoring how good they are

**Our optimal configuration ($\tau_c = 0.05$):**

- Cost matters, but performance remains the primary consideration
- We achieve the optimal balance between capability and efficiency

### B.6.3 Judge Weighting ($\tau = 300$)

**The Question:** How much more should better judges influence decisions?

**If we weight too little ($\tau = 100$):**

- Top judges dominate 90% of votes
- We lose diversity of opinions

**If we weight too much ($\tau = 500$):**

- All judges have nearly equal weight
- We lose the benefit of better judges
- Agreement drops to 72%

**Our optimal configuration ($\tau = 300$):**

- Achieves an appropriate balance between judge quality and diversity
- Achieves 89% agreement between judges

### B.6.4 MATCH FAIRNESS (±50 ELO)

**The Question:** How close in skill should opponents be?

**If too strict (±25 Elo):**

- 15% of models can't find opponents
- System breaks down

**If too loose (±100 Elo):**

- Mismatches with 85% win rates
- Ratings become inaccurate (±45 points)

**Our optimal configuration (±50 Elo):**

- Ensures competitive matches for all participants
- Achieves accurate ratings (±18 points)

## B.7 HOW RELIABLE ARE OUR RESULTS?

### B.7.1 RATING ACCURACY

**The Question:** How confident can we be in each model's rating?

**Our Answer:** 95% of models have ratings accurate to within ±18 points after 25-30 games

**Confidence Interval Calculation:**

$$\text{CI} = \pm 1.96 \times \frac{\sigma}{\sqrt{n}} \tag{28}$$

Where:

- $\sigma$ is the rating standard deviation
- $n$ is the number of matches
- Target: ±18 Elo with 25-30 matches

**What this means:**

- If a model is rated 1600, we're 95% sure their true rating is between 1582-1618
- This is much more accurate than other systems (±35 points)
- We need fewer games to get reliable results

### B.7.2 HOW FAST DO RATINGS STABILIZE?

**The Process:**

1. Start with all models at 1500 rating
2. Play games and update ratings
3. Stop when ratings stop changing much

**Our Results:**

- Most models stabilize after 25-30 games
- Ratings stop bouncing around
- We can trust the final rankings

### B.7.3 How Much Do Judges Agree?

**The Question:** When multiple judges score the same match, do they agree?

**Agreement Rate Calculation:**

$$\text{Agreement} = \frac{\sum(\text{Consistent\_Judgments})}{\sum(\text{Total\_Judgments})} \tag{29}$$

$$\text{Weighted\_Agreement} = \frac{\sum(w_k \times \text{Agreement}_k)}{\sum(w_k)} \tag{30}$$

**Our Results:**

- Our AI judges agree 89% of the time
- Human judges only agree 72% of the time
- Our system is more consistent than humans!

**Why this matters:**

- Consistent judging = reliable rankings
- Less random variation in results
- More trustworthy evaluation

## B.8 How We Stop Cheating

### B.8.1 The Cheating Problem

Some models might try to game the system by:

- Coordinating with other models to give each other good scores
- Being biased toward models from the same company
- Manipulating the judging process

### B.8.2 Our Anti-Cheating Measures

**Random Audits:**

- 5% of all matches are randomly checked by humans
- Creates a deterrent against systematic cheating
- Maintains efficiency while ensuring quality

**Pattern Detection:**

- We watch for unusual voting patterns
- Detect if judges from the same company always agree
- Spot if certain models always get favorable treatment

**Cryptographic Security:**

- All decisions are cryptographically signed
- Impossible to tamper with results after the fact
- Complete audit trail of every decision

**Cross-Provider Validation:**

- Judges from different companies evaluate the same matches
- If they disagree too much (30%+), we investigate further
- Prevents single-company bias

### B.8.3 WHY THIS WORKS

**Multiple Layers of Protection:**

- Even if someone cheats one way, other measures catch them
- Random audits make systematic cheating risky
- Cryptographic signatures make tampering impossible
- Cross-validation prevents bias

## B.9 THE REAL-WORLD IMPACT: WHY COST MATTERS

### B.9.1 THE HIDDEN COST PROBLEM

Traditional benchmarks ignore cost, but in the real world, cost is everything. Consider a company processing 1 million queries per month:

**Naive Approach (Use only the best model):**

- Use GPT-4.1 for everything
- Cost: $226,500 per month
- Performance: 100%

**Smart Approach (Use our tiered system):**

- Critical tasks (5%): Use Gemini 2.5 Pro - $11,325
- Important tasks (20%): Use GPT-4.1 mini - $1,332
- Standard tasks (75%): Use Qwen 3.2 235B - $266
- Total Cost: $12,923 per month
- Performance: 98.2%

**Savings: 94.3% cost reduction with only 1.8% performance loss!**

### B.9.2 HOW WE CALCULATE EFFICIENCY

**The Formula:**

$$\text{Efficiency} = \frac{\text{Performance}}{\text{Cost}} \tag{31}$$

**Example:**

- Model A: 1600 rating, costs $10 $\rightarrow$ Efficiency = 160
- Model B: 1550 rating, costs $1 $\rightarrow$ Efficiency = 1550
- Model B is 9.7× more efficient despite lower performance!

### B.9.3 THE 641× EFFICIENCY GAP

Our system revealed massive efficiency differences invisible to traditional benchmarks:

**The Discovery:**

- Qwen 3.2 235B: 97% of top performance at 0.16% of the cost
- This represents a 641× efficiency difference between similar-performing models
- Traditional benchmarks completely miss this!

**Why This Matters:**

- Companies can save millions by choosing efficient models
- Small performance differences often aren't worth huge cost increases
- Real-world deployment decisions should consider both performance AND cost

## B.10 How Fast Do We Get Reliable Results?

### B.10.1 The Convergence Process

**What Happens:**

1. Start with all models at 1500 rating
2. Play games and update ratings after each match
3. Watch ratings change and stabilize
4. Stop when ratings stop changing much

**The Math:**

- Each game changes ratings by: $|\Delta R| = K \times |S - E|$
- If a model wins when expected to lose: rating goes up a lot
- If a model loses when expected to win: rating goes down a lot
- If result matches expectation: rating changes little

**Convergence Rate:**

$$\text{Convergence\_Rate} = 1 - \frac{|\Delta R|}{|R|} \tag{32}$$

$$\text{Stability} = 1 - \frac{\sigma_{\text{rating}}}{\sigma_{\text{initial}}} \tag{33}$$

### B.10.2 How Many Games Do We Need?

**Our Results:**

- Most models stabilize after 25-30 games
- Ratings become accurate to within ±18 points
- This is much faster than other systems (which need 80+ games)

**Why This Matters:**

- Faster convergence = less time and money spent
- More accurate ratings = better decisions
- Reliable results sooner = practical deployment

### B.10.3 When Can We Trust the Rankings?

**The Criteria:**

1. Ratings stop changing much between games
2. 95% of models have ratings accurate to ±18 points
3. Judge agreement is above 89%

**Our Achievement:**

- All criteria met after 25-30 games
- Much more reliable than human evaluation (72% agreement)
- 2× more accurate than existing systems

### B.11 CAN THIS SCALE TO REAL COMPANIES?

#### B.11.1 HOW MUCH COMPUTING POWER DO WE NEED?

**The Good News:**

- Each game only needs simple math (O(1) time)
- Rating updates are instant
- Judge weighting is fast
- Swiss pairing is efficient (O(N log N))

**Real Example:**

- We ran 48 models on a regular laptop (M3 MacBook, 16GB RAM)
- Total cost: $245.65 for all evaluations
- No special hardware needed!

#### B.11.2 WHAT ABOUT 1000+ MODELS?

**Scaling Up:**

- More models = more matches needed
- But the math stays the same
- Can run matches in parallel
- Memory usage grows linearly with models

**Optimization Strategies:**

- Run independent matches simultaneously
- Cache frequently used ratings
- Update ratings incrementally
- Use distributed computing for huge deployments

### B.12 HOW DO WE KNOW THIS ACTUALLY WORKS?

#### B.12.1 THE HIERARCHY TEST

**The Question:** Do our rankings match what we expect from model generations?

**Our Test:** Check if newer models consistently beat older models from the same company

**Results:**

- OpenAI: GPT-4.1 (1601) ¿ GPT-4o (1515) ¿ GPT-3.5 (1414)
- Anthropic: Claude 3.7 (1531) ¿ 3.5 (1510) ¿ 3.0 (1471)
- Google: Gemini 2.5 Pro (1604) ¿ 2.0 Flash (1520) ¿ 1.5 Pro (1457)

**Why This Matters:** Our system correctly identifies that newer models are better, proving it works!

#### B.12.2 CROSS-DOMAIN VALIDATION

**The Test:** Do models that perform well in one area also perform well in others?

**Results:**

- Programming vs Mathematical Reasoning: 78% correlation
- Clinical vs Programming: 72% correlation
- Strong correlation shows our ratings capture real capability

### B.12.3 PARAMETER SENSITIVITY

**The Question:** What if we change our parameters slightly?

**Our Findings:**

- Small changes in parameters don't break the system
- Rankings stay mostly the same
- System is robust and reliable

### B.13 SUMMARY: WHY THIS MATH MATTERS

**The Big Picture:**

1. **Dual-Track Ratings** show both performance AND cost efficiency
2. **Swiss Pairing** ensures fair evaluation for all models
3. **Smart Judge Weighting** balances quality and diversity
4. **Cost Integration** reveals 641× efficiency differences
5. **Anti-Gaming** prevents manipulation and bias
6. **Fast Convergence** gives reliable results in 25-30 games
7. **Real-World Impact** enables 94% cost savings with minimal performance loss

**The Bottom Line:** This isn't just academic math - it's a practical system that helps companies make better decisions about which AI models to use, saving millions while maintaining performance.

