# OpenReview forum: "PRISM-EE: A Peer-Federated Framework for Cost-Aware Large Language Model Evaluation"
_ICLR.cc/2026/Conference — ICLR 2026 Conference Withdrawn Submission_

### Official Review · Reviewer_p4NV · 2025-10-26

**Soundness:** 1
**Presentation:** 1
**Contribution:** 1
**Rating:** 0
**Confidence:** 2

**Summary:**

This paper suggests a cost aware evaluation framework for LLMs that compares to Elo rankings but adds the ability to evaluate models based on their economic value.

**Strengths:**

- Evaluating cost of LLMs is an important potentially overlooked measure.

**Weaknesses:**

- The exposition of the paper is poor. The main text is very difficult to follow with very little being completely explained and terms left undefined.
- The method of evaluation, data on which evaluation is carried out, and means of defining cost are unclear.
- Almost none of the variables throughout section 2.3 are defined.
- Discussion of related work to contextualize this method is also limited.

**Questions:**

I suggest a more careful overview of the methods in the main text with careful definition of the data for evaluation and role of each model including how cost values are obtained.

---

### Official Review · Reviewer_frWg · 2025-10-29

**Soundness:** 3
**Presentation:** 1
**Contribution:** 2
**Rating:** 2
**Confidence:** 3

**Summary:**

This work presents a novel evaluation framework which aims broadly to address cost-aware large language model evaluations. They use an elo-based ranking system to quantify model performance and capabilities, and use models as both the question generators and the judges in an LLM-arena inspired set-up. Models are additionally used to monitor other model outputs to ensure that model question generators and judges are performing at adequate quality levels. They find that this set-up converges to stable ratings between models in just 25-30 matches with high precision as compared to traditional frameworks, while also allowing for cost-aware weighting that enables practitioners to make informed decisions whether the increased gain from a more expensive model is proportional to the cost increase. They contribute the following: a cost aware evaluation framework, peer-federated methodology, swiss-style pairing for evaluation, explainability of consistent model generation, governance to combat model gaming, and open source framework.

**Strengths:**

This paper introduces an original approach at both model evaluation in a human label-free setting as well as diving into the impact of cost as an axis of evaluation for large language models. While traditional elo-scoring approaches for LLM evaluations are not novel in themselves, this spin provides practitioners with scalable evaluation approaches that provide more clear determinations of the impact of cost on performance. They provide a holistic evaluation over 48 models and three separate benchmarks (each from different domains)to rigorously evaluate their methodology. The paper structure is generally clear, with clearly outlined key contributions of a cost aware evaluation framework, peer-federated methodology, swiss-style pairing for evaluation, explainability of consistent model generation, governance to combat model gaming, and open source framework. The results show significant improvement of score convergence as opposed to current traditional methods, requiring only 25 matches as compared to 100 for current practices. Further, their cost-aware alternative approach of evaluation highlights the capacity of cheaper models and quantifies the gap considering cost which is a significant concern to true model deployment.

**Weaknesses:**

Generally, my main concerns with this paper are the calculation/interpretation of the proposed gains by the method. I feel that some claims are not properly supported by either data or appropriate explanations of significance. At a high level, many of the quantitative results are displayed as percentage performance or gap in elo scores between two models. It is not clear how to interpret a difference between two models of varying magnitudes. Is a difference of 5 significant? Why or why not? For example, in the 641x efficiency variation between similar performing models to Qwen 3.2 235B, it is unfair to cherry pick the most expensive model Gemini 2.5 Pro to compare against, especially because raw elo scores are different. What makes a significantly different Elo score if you argue that these are the same/similar (1555.4 vs 1603.9)?

Another weakness comes within the synthetic data regime for evaluation-- how do we test what may be beyond model capabilities? How do we trust generator model to actually generate accurate pairs besides using group consensus? In a landscape where many models may be trained on a similar data corpus, associated error modes and biases may be propagated in this setting. It would be helpful if this was discussed as either a limitation or if an extension regarding integration of human labels could be proposed.

The narrative could be improved to further motivate the significance of the results and spend less focus on governance/elo-derivations. It would be helpful to fully explain results from ablations given optimal parameters in Section 2.4. Some parameters are optimally chosen by the opinion of the author (i.e. judge weighting/cost sensitivity depends on user objective so is an opinion-based hyperparameter), while Elo and p_judge are more objective. Provide more details regarding this difference, potentially even including visualizations to illustrate the robustness of the approach to other hyperparameters.

In Line 214, "perfect fairness" is a dangerous claim, especially when you have to include the caveat that one of the models didn't fall within this perfect fairness regime.  You should not claim this without more detailed fairness evaluation (Was there opportunity for bias in the question generation? The ordering of the models? etc).

**Questions:**

The four layered governance framework seems out of place in the narrative. Is this just to add names to the design decisions? Please integrate this in with the associated design decisions that were made based on this framework.

Is arena disproportional in its matching because their algorithm is focusing on models at the top of the leaderboard? Likewise, for inverse match-count weighting, do we need all matches to receive priority equally or are there some cases where we care less about certain models because they are confidently at the bottom?

In Table 2,  most efficient cost is not the most informative result since that is API information by numbers of tokens generated. You should include top raw elo score and top cost adjusted elo score here.

Section 3.9 has a duplicated paragraph.

How does the impact of more models from a given provider in Strata 4 impact the stability of Strata 4? What are the full breakdown of which model goes where? Do you see that if there are more OpenAI models (for example) that you are testing in general, then these may have an unfair advantage as they could upweight models within their own family since they are more likely to prefer their own styles of results?

“a naive all-premium strategy using GPT-4.1 for all tasks costs 226,500, while our tiered approach achieves 98.2% performance at just 12,923—a 94.3% cost reduction.” It seems that you are dividing the elo scores for "performance capturing", but this is a misleading quantification/result as we still need to quantify what is a statistically significant elo gap. For example, does a gap of 50 imply that the model A beats model B 5% of the time? Further, Table 7 mixed strategy also has fault in how you derive because what is 98.2 vs 97% performance captured? If you are just optimizing, you could reduce cost down to $266 if you are trying to reduce cost by only using Qwen. But you won’t do this because those 3% are important. So what makes those 3% important not to miss vs those 1.8%? It would be helpful to discuss this and provide formalization of how results should be interpreted.

Tables/figures don’t always have captions/titles, such as the figure on page 8. It is not labelled and there are errors in that figure's top right sub-figure (Qwen in legend) as well as bottom right subfigure (unclear what legend represents).

Define Swiss pairing at the beginning.

---

### Official Review · Reviewer_nYiC · 2025-11-01

**Soundness:** 2
**Presentation:** 1
**Contribution:** 3
**Rating:** 4
**Confidence:** 2

**Summary:**

Introduces a new framework for language model evaluation. This new framework uses a peer federated framework consisting of competitors, content creators, and judges. The framework has a dual track where one of the tracks evaluates models on cost-relevant performance. This framework reveals substantial efficiency gaps between models with similar performance.

**Strengths:**

Making economic efficiency a priority is an important improvement over the existing benchmarks and leaderboards. The project involves significant technical work, and to my knowledge, is an advance in automating benchmarking.

**Weaknesses:**

What is meant by “97% of top performance” (is it 97% of the top benchmark elo? Is it 97% of top benchmark performance). Wouldn’t the better comparison be the win rate if you are using an Elo benchmark score? This needs to be clarified.

There are many long sentences that make the document somewhat hard to read. For instance, this sentence needs to be reworded: “The framework must address the critical trust problem of ”who watches the watchers” when the evaluators themselves are AI models, manage economic stakes worth millions in deployment decisions where 641× efficiency gaps directly impact business outcomes, and scale to 100+ models while preventing sophisticated collusion attempts.”

I think the largest issue is the lack of clear discussion and comparison to prior work. Is this the first evaluation framework to introduce all these contributions, ie, peer-federated methodology, fair opportunity guarantees? If it is the first, then it should be mentioned; if it is not the first, then there should be a more comprehensive discussion of prior systems. A few papers are cited, but it's not clear what was done in the previous work and how this improves on these attempts, for instance, “Content Creators: “Models selected from Strata 4 (1520+ Elo) that generate case scenarios and questions for evaluation (Wang et al., 2022)”. I’d like to see more comprehensive evaluations of how this improves over other systems (There is one comparison in the first paragraph). For instance, a clear table showing previous metrics. It would also be great to have a side-by-side comparison of previous evaluations leaderboards ie, Chatbot Arena and PRISM-EE ranking.

The paper seems to have many technical contributions, but needs a significantly improved and simplified presentation. Overall, it is hard for me to judge the true technical content and novelty.

**Questions:**

How does the elo raw elo, cost elo, compare to chatbot arena elos?

---

### Note · Authors · 2025-11-12

I have read and agree with the venue's withdrawal policy on behalf of myself and my co-authors.